# Synergistic Effect of Activated Carbon, NiO and Al_2_O_3_ on Improving the Thermal Stability and Flame Retardancy of Polypropylene Composites

**DOI:** 10.3390/polym15092135

**Published:** 2023-04-29

**Authors:** Mingqiang Shao, Ying Li, Yiran Shi, Jiangtao Liu, Baoxia Xue, Mei Niu

**Affiliations:** 1College of Textile Engineering, Taiyuan University of Technology, Taiyuan 030600, China; 2College of Chemical Engineering and Technology, Taiyuan University of Technology, Taiyuan 030024, China

**Keywords:** polypropylene composites, activated carbon, NiO, Al_2_O_3_, carbon nanotubes, flame retardancy, thermal stability

## Abstract

It is difficult to enhance the char yields of polypropylene (PP) due to the preferential complete combustion. Successful formation of abundant char layer structure of PP upon flammability was obtained due to the synergistic effect of NiO, Al_2_O_3_ and activated carbon (AC). From characterization of scanning electron microscopy (SEM) and transmission electron microscopy (TEM), it was revealed that the microstructure of residual char contained large amount of carbon nanotubes. Compared to the modification of AC, NiO and Al_2_O_3_ alone, the combination of AC, NiO and Al_2_O_3_ dramatically promotes the charring ability of PP. In the case of AC and NiO, NiO plays a role of dehydrogenation, resulting in the degradation product, while AC mainly acts as carbonization promoter. The addition of Al_2_O_3_ results in higher dispersion and smaller particle size of NiO, leading to greater exposure of active sites of NiO and higher dehydrogenation and carbonization activity. Compared to the neat PP, the decomposition temperature of the PP modified by combined AC, NiO and Al_2_O_3_ was increased by 90 ℃. The yield of residual char of AC-5Ni-Al-PP reached as high as 44.6%. From the cone calorimeter test, the heat release rate per unit area (HRR) and total heat release per unit area (THR) of PP composite follows the order AC-5Ni-Al-PP < AC-10Ni-Al-PP < AC-Ni-PP < AC-15Ni-Al-PP < AC-1Ni-Al-PP. Compared to the neat PP, the peak of HRR declined by 73.8%, 72.7%, 71.3%, 67.6% and 62.5%, respectively.

## 1. Introduction

As one of the three general-purpose resins, PP is widely used in many areas of life, including electrical appliances, automobiles, fibers, furniture and so on, due to its good mechanical strength, insulation property and corrosion resistance [1,2]. However, PP material has an inherent drawback. The Limiting Oxygen Index of PP is nearly 17% and no carbonization residue exists after combustion, indicating the flammability of the material. Furthermore, the combustion of PP is violent and quick, restricting the more widespread application considering the safety. As a result, the improvement of flame retardancy is a major problem in the application of PP material.

It is an important method to enhance the flame retardancy of PP by improving the charring ability of combustion products [3,4,5]. There are various ways of promoting the flame retardancy, including magnesium–aluminum inorganic flame retardants, phosphorus flame retardants, intumescent flame retardants and other types [6,7,8]. Especially, the intumescent flame retardants display good effects, forming an effective and protective carbon layer [8,9]. It is a flame retardant system that forms a porous expanded carbon layer during the heating process and promotes the flame retardant effect by insulating the heat and oxygen and preventing the volatilization of pyrolysis products [10]. Aside from the compact carbon layer, the type of carbon nanotubes (CNTs), carbon nanosheet and carbon dots have also received much attention [11,12,13,14,15]. In the case of CNTs with high ratio of length to diameter, the continuous network structured protective layer forms a barrier that blocks the heat transfer and volatilization of combustible organics. Jiangbo Wang reported the improved thermal stability of epoxy resin by 2%wt CNTs, resulting in a 30.5% reduction in peak heat release rate (PHRR) due to the effective reduction in heat release and transfer [16]. Yan Zhang et.al reported that the heat and flammable gases decrease, introducing CNTs into poly(butylene succinate). Furthermore, the yields of toxic gases of CO and CO_2_ are dramatically reduced by 50% or so [17].

Aside from the conventional CNTs with the polymers by blending, in situ formation of CNTs during combustion is proven to obtain the effective flame retardancy [18,19,20]. Synthesis methods of CNTs have been widely researched. In general, CNTs were abundantly synthesized with monometallic or bi-/trimetallic components used as catalysts and methane, ethylene or other hydrocarbons acting as carbon sources [21]. Recently, a new method of “combined catalyst” was proposed to synthesize CNTs in which polymer was used as a carbon source rather than the small molecules. In the system, metals such as Fe and nickel components catalyze the polymer into dehydrogenation products and then CNTs are formed under the action of a carbonization promoter, such as SiO_2_, MgO or activated carbon [19,22,23,24,25]. Tang et.al explored the composite that fumed silica and Ni_2_O_3_ incorporated into PP in which the CNTs and carbon nanofibers were produced and the char residual was raised from 0.2% to 18.2% [24]. A dehydrogenation reaction mechanism includes free radical mechanism and ionic mechanism. The catalysts include metals, metal oxides, sulfides, phosphides and carbides, such as pt, pd, NiO_x_, Nb_2_O_3_, WS_2_, MoS_2_, Ni_2_P, MoC and so on [26,27,28,29]. Among them, noble metals pt and pd show high hydro/dehydrogenation ability at reaction temperatures as low as 180 °C [30]. However, the combustion temperature is far beyond 180 ℃ and can even reach as high as 700 °C, which leads to severe aggregation and inactivation of metallic particles. Compared to noble metal, NiO_x_ sustains high hydro/dehydrogenation ability at high reaction temperature, and the particle is not easy to aggregate [31]. Further, the NiO_x_ is also cheap, which is beneficial for facile large-scale utilization in the field of flame retardants. Compared to MgO, SiO_2_ or other carbonization promoters, AC is not only low-cost but also enhances the cracking efficiency of PP into small molecules [32]. Further, synergism of Ni and AC also promotes the catalytic dehydrogenation and nanosheet growth. Al_2_O_3_ also acts as an additive, which could promote the crosslinking of the dehydrogenation products and form the carbonization film, preventing the feedback of heat. In addition, the Al_2_O_3_ could also interact with NiO_x_, enhancing the dispersion of NiO_x_ particles [23].

Therefore, in this study, a novel flame retardant system combined with NiO, Al_2_O_3_ and AC was facilely synthesized. It is proven that the combined AC/NiO/Al_2_O_3_ catalyst could catalyze PP into a large amount of CNTs, promoting the thermal stability and flame retardancy. The reactants and residual char after combustion were analyzed with different means to study the degradation behavior, combustion behavior and mechanism of forming the CNTs.

## 2. Materials and Methods

### 2.1. Materials

First, 30 mesh of PP was a commercial product (Mw = 29.7 × 10^4^ g/mol, Polydispersity = 3.68, melting point = 166.9 ± 0.5 °C, melt flow rate = 4 ± 0.25 g/10 min, melt flow index (2.16 kg/190 °C) = 11.2 ± 0.29 g/10 min) (Huachuang Chemical Co., Ltd. Shanghai, China). Further, 200 mesh of activated carbon, commercial Ni(NO_3_)_2_·6H_2_O and Al(NO_3_)_3_·9H_2_O and polyethylene glycol were purchased from Aladdin Biochemical Technology Co., Ltd. (Shanghai, China).

### 2.2. Preparation of PP Composites

Firstly, designed amounts of Ni(NO_3_)_2_·6H_2_O and Al(NO_3_)_3_·9H_2_O were dissolved polyethylene glycol and mixed homogeneously and then calcined at 650 °C for 30 min. Subsequently, PP composites were obtained by mixing PP with a ratio of resultant NiO, Al_2_O_3_ and AC. A series of PP composites were synthesized and their detailed compositions were listed in Table 1. The resultant composites were extruded into a sample with a size of 10 × 10 × 3 cm^3^ by a curing press.

### 2.3. Characterization

The phase structures of PP composites before and after combustion were both analyzed by X-ray diffraction (XRD) using TD-3500 diffractometer with Cu Ka radiation at λ = 1.542 Å, providing a voltage of 40 kV and 40 mA. It is noted that the samples after combustion were analyzed without any purification by HF or HNO_3_.

H_2_ temperature programmed reduction (H_2_-TPR) experiments were carried out on a TP-5080B instrument equipped with a TCD detector. A sample (≈100 mg) was pretreated in a nitrogen atmosphere at 300 °C for 1 h. After that, nitrogen was exchanged into 10 vol% H_2_ in nitrogen with a constant flow rate of 30 mL min^−1^ and the sample was cooled to 40 °C. Finally, the sample was heated to 750 °C at a rate of 10 °C/min.

The Raman spectrometer was performed on a SPEX-1403 to study the purity and graphitization degree of the produced carbon, and the spectra were collected at a laser wavelength of 532 nm with Raman shift from 1000 to 2000 cm^− 1^.

Thermogravimetry (TGA) for PP and its composites was completed using a thermal analysis instrument (TG4000, PerkinElmer) from room temperature to 700 °C in air and nitrogen atmosphere with a heating rate of 10 °C/min. The sample measured is 10 mg.

Imitative combustion tests were performed to test the residual char. Firstly, a piece of sample (4.0 g ± 0.1 g, 4 cm × 2 cm × 0.3 cm) compressed by curing press was placed in a short quartz tube and the tube was pushed to the middle of tubular furnace, which was heated at 700 ℃, and then the residual char was collected and weighted.

Cone calorimeter tests were conducted using a Fire Testing Technology Ltd. (West Sussex, UK) according to ISO 5660-1 standard method with an external heat flux of 50 kW m^−2^. The size and thickness of each specimen were 100 mm × 100 mm × 6 mm square plaques, molding at 180 °C by vulcanizing machine.

Morphologies and microstructure of residual chars of PP composites were observed by means of TEM (FEI Tecnai G2 F20 and JEM-F200) at 200 kV accelerating voltage. The analyzed samples were placed in ethanol with an ultrasonic dispersion for an hour and deposited on a Cu grid, and then the samples were dried overnight. It is noted that the samples after combustion were analyzed without any purification by HF or HNO_3_.

The morphologies of the typical samples were recorded by scanning electron micros copy (SEM) (Sigma500, ZEISS, Jena, Germany) at 5 kV accelerating voltage.

## 3. Results and Discussion

First, compositions of PP composites before and after combustion were investigated by XRD. In addition, the reduction property of PP composites was also investigated by H2-TPR. The results are presented in Section 3.1 and Section 3.2, respectively. Next, the thermal stability and residual char morphology of PP composites were investigated in detail, which are presented in Section 3.3 and Section 3.4. Then, the effects of the combined AC/NiO/Al_2_O_3_ catalyst on the flammability of PP composites were discussed from the cone calorimeter tests, and the results are presented in Section 3.5.

### 3.1. Composition Analysis of PP Composites before and after Combustion

To investigate the change in PP composites before and after combustion, wide-angle XRD analysis was performed. The results are presented in Figure 1. The characteristic peaks of PP (2θ = 14.2°, 17.0°, 18.6°, 21.8°) were observed for the synthesized samples and completely disappeared after suffering from heating treatment at 700 °C in air [33]. For the composites containing Ni or Al elements, the diffraction peaks at 40.2°, 46.8° were ascribed to the NiO from (111) and (200) planes (JCPDS No. 04–0850). For the NiO component, it is noted that the intensity declines and the full width at half maxima increases with increasing content of Al, indicating that the crystalline particles reduce and the dispersion increases. After combustion, the characteristic peaks of Ni (48.1°, 56.1°) were observed, while the peaks of NiO were weakened, suggesting that part of NiO was transformed into Ni. During the combustion, the PP degrades into H_2_ and other hydrocarbons, such as CO, CH_4_, olefins and alkynes, in which the H_2_ and CO in situ reduce the NiO into Ni. Furthermore, it is also found that two new peaks at 26.3°, 42.6°appeared, which is ascribed to the graphite characteristic peaks of (002) and (101).

Raman spectroscopy was used to study the graphitization degree of the combustible products according to the vibrational modes. Among them, the D-band at 1350 cm^−1^ represents the impurities in sidewall structure and the G-band is associated with the graphene layers order of carbon nanotubes [34]. Therefore, the higher the ratio of I_G_ to I_D_, the higher the graphitization degree. The Raman spectrum and ratio of I_G_ to I_D_ were presented in Figure 2. It is observed that the I_G_ to I_D_ ratio was in the range of 0.662 to 0.754 over the combustible products, and the value first increased and then decreased with increasing amount of Al, in which the AC-5Ni-Al-PP shows the highest value and graphitization degree. Furthermore, it was also found that the ratio of I_G_ to I_D_ over AC-1Ni-Al-PP is lower than AC-Ni-PP, suggesting that the excessive introduction of Al amount is not beneficial for the enhancement of graphitization degree.

### 3.2. Reduction Property of Synthesized PP Composites

Due to CNTs being formed under the action of metallic Ni reduced from NiO, characterization of H_2_-TPR was carried out to investigate the reducibility of the prepared samples (Figure 3). It is known that, the higher the temperature corresponding to the peak, the more difficult it is to be reduced. The broad peaks between 200 and 500 °C are assigned to reduction of NiO [35]. It is found that the reduction peak of NiO assigned to Ni^2+^ is sharp and the corresponding temperature is 355 °C. After introduction of Al, the curves become less steep. Furthermore, because of the interaction between NiO and Al_2_O_3_, all of them shift toward higher temperatures from 365 to 406 °C with increasing Al amount. It is also found that there is a small peak at about 926 °C for 10Ni-1Al, 5Ni-1Al and 1Ni-1Al, which is assigned to the reduction of NiAl_2_O_4_ spinel [36]

### 3.3. Effect of the Combined AC/NiO/Al_2_O_3_ Catalyst on the Thermal Stability and Yields of Residual Char

The thermal stability measurements of PP and its composites in air and nitrogen atmosphere were conducted. The TG and DTG results were presented in Figure 4. Furthermore, the onset decomposition temperature (T_5_), the 10% decomposition temperature (T_10_), the half decomposition temperature (T_50_) and the maximum decomposition temperature (T_max_) were listed in Table 2 and Table 3, respectively. From the results obtained in air, it was observed that the T_5_, T_10_, T_50_ and T_max_ of pure PP were 297.1, 315.2 and 382.1 °C, respectively. After introduction of individual or combined catalyst, the corresponding temperatures of T_5_, T_10_, T_50_ and T_max_ all remarkablely enhanced. Among them, the AC-PP increased by 64.8, 72.1, 66.8 and 75.4 °C, respectively. This is attributed to the fact that the addition of AC into the PP may promote the curing and the cross-linking degree of PP [16]. When only NiO was added into PP, the T_5_, T_10_, T_50_ and T_max_ increased by 48.2, 49.4, 44.8 and 47.6 °C, respectively. However, when both NiO and AC were used, the value increased by 72.2, 84.2, 73 and 76.1 °C, respectively. This indicates that the addition of both NiO and AC has a synergistic effect and plays a more effective role for promoting thermal stability than individual NiO or AC. In addition, it is also observed that the introduction of Al_2_O_3_ into the system enhanced the decomposition temperature overall. The AC-5Ni-Al-PP dramatically promotes the thermal stability and the T_5_, T_10_, T_50_ and T_max_ increased by 85, 91.3, 94 and 90.6 °C, respectively. The result shows that too much or too little addition of Al_2_O_3_ is not beneficial for the thermal stability. For the samples measured in nitrogen atmosphere, the results show similar change, but the improvement in thermal stability is not more significant than that measured in air atmosphere.

Imitative combustion experiments were conducted and all the samples were re- peated three times for reproducibility. The results are presented in Table 1. For the pure PP, there is almost no residual char left. It is similar to the neat PP that there are slight yields of residual char for Ni-PP and Al-PP. In the case of AC-PP, the residual mass is even lower than the initial addition amount of AC. There is no obvious increase in residual mass of AC-PP, Ni-PP and Al-PP, indicating that the individual component of AC, NiO and Al_2_O_3_ could not effectively catalyze carbonization of the pyrolytic products of PP during combustion. In contrast, when binary or ternary composites (NiO, AC or Al_2_O_3_) were simultaneously used, the residual mass was dramatically increased from 24.8 to 44.6%. Therefore, there is a strong synergistic effect catalyzing the degradation products into char when the AC, NiO and Al_2_O_3_ coexists. From the SEM and TEM results, it is proven that the char remained in the form of CNTs (see Figure 5 and Figure 6).

### 3.4. Morphology and Microstructure Analysis of Residual Char

Morphology of typical composites after cone calorimeter tests is presented in Figure 5a–c. For the neat PP sample, no obvious residual char is visible due to the easily complete combustion. Small amounts of residual char (11.3 wt%) were observed from AC-PP, which is lower than that of the initial addition of 15 wt%. This is in accordance with the imitative combustion measurement. When the NiO or NiO/Al_2_O_3_ were added, large amounts of residual char were retained, which reached as high as about 40 wt%. In comparison with AC-Ni-PP, more residual char was produced from the AC-5Ni-Al-PP. SEM was used to investigate the microstructure of typical products after combustion and the results are presented in Figure 5d–f. For the AC-PP, different sizes of AC aggregate and create a variety of holes. After addition of NiO, the microstructure of AC-Ni-PP after combustion is totally different. A large amount of CNTs are observed and part is closely intertwined. This easy catalytic combustion method is allowable for large-scale production of CNTs. Unfortunately, the quality of CNTs is not excellent in that the CNTs surface is unsmooth and many bulges exist, which is also confirmed by the TEM. It is speculated that the combustion is violent and the reaction components are complex, resulting in the non-uniformity of CNTs growth. For the AC-5Ni-Al-PP, the abundant CNTs are also observed. In comparison with AC-Ni-PP, the proportion of winding CNTs is higher and intertwined phenomenon is also more severe, which is in accordance with the previous report [23]. It is interesting that only AC was added into the PP; the residual char still contains a great deal of carbon. While AC, NiO and Al_2_O_3_ were added, no abundant carbon pieces were observed. This interesting phenomenon will be further researched in the future work.

TEM was conducted to obtain the information regarding the microstructure of the residual char of PP composites after imitative combustion experiment (Figure 6). It is obvious that the PP modified by AC, NiO or Al_2_O_3_ alone shows irregular shape. The amorphous structure is mainly composed of the additional material. For the Ni-PP, only a slight amount of carbon material was observed except for the NiO, indicating that the individual addition could not significantly promote the production of residual, just as in the form of carbon. In contrast, a large number of products were observed in the form of carbon nanotubes for the multicomponent modification PP composites. Overall, for the different samples, there are some differences that the diameter and length of CNTs are nonuniform. Especially for the AC-Ni-PP and AC-15Ni-Al-PP, the size distribution of CNTs is wide, while, for the AC-10Ni-Al-PP, AC-5Ni-Al-PP and AC-10Ni-Al-PP, the CNTs diameter is smaller and relatively uniform in comparison with the AC-Ni-PP and AC-15Ni-Al-PP. Due to the interaction between NiO and Al_2_O_3_, the dimension decreases with increasing the Al_2_O_3_ addition, which is in accordance with the XRD results. It is concluded that aggregation of NiO nanoparticles happens easily when only NiO, AC or small amounts of Al_2_O_3_ exist, resulting in a large difference in diameter. With increasing the Al_2_O_3_ amounts, the phenomenon of aggregation dramatically declines and dispersion of NiO nanoparticles increases, enabling the smaller size and uniform distribution of Ni nanoparticles. Furthermore, from Figure 6i, it is obviously observed that the metallic particle is wrapped by CNTs. It is interesting that some CNTs contain several Ni nanoparticles, while part of the CNTs only contains one particle at the top. At the same time, it is also observed that the dimension of CNTs is irregular and coarse over the surface, which is consistent with the SEM result. For the PP composites that contain high content of Al_2_O_3_, such as AC-1Ni-Al-PP, the ratio of length to diameter of CNTs is high due to the high dispersion and small NiO particles. Usually, the higher the ratio of length to diameter, the more compacted the network structure protective layer preventing the heat transfer and volatilization of degradation products [20]. However, due to the high reduction difficulty, the yields of residual char for PP composites with high content of Al_2_O_3_ are low. Finally, the flame retardancy effect is not ideal (see Figure 7) owning to a compromise between quality and quantity of CNTs. From Figure 6I, the D-spacing is 0.34 nm, which was consistent with the ideal graphitic interlayer spacing. The growth direction of graphene layers is also parallel to the CNT axis. The metallic particle after combustion was further analyzed and it was found that the D-spacing value of the particle is 0.2088 nm, corresponding to the NiO (200) (see red arrow in Figure 6I). However, the D-spacing value is 0.2031 nm over the other particle, corresponding to the Ni (100) (see yellow arrow in Figure 6II). This result is in accordance with the XRD results that Ni and NiO all coexisted. Furthermore, it also evidences that NiO is transformed into Ni during combustion, which participates in the production of CNTs rather than raw NiO. We also found that NiO is still dominant compared to Ni in the char. It is speculated that the reduction condition results in this phenomenon. On the base of H_2_-TPR measurements, NiO is completely reduced into Ni with the time of about 20 min at a flow of 30 mL/min, 10 vol% H_2_/Ar. However, the concentration of the reducing atmosphere, such as H_2_, CO or CH_4_ produced during combustion, is low, which cannot transform the NiO into Ni completely.

### 3.5. Effect and Mechanism Analysis of the Combined AC/NiO/Al_2_O_3_ Catalyst on the Flammability of PP

It is well known that the cone calorimeter test acted as an effective means and has been used to research the combustion property [37]. To investigate the combustible difference in various samples and reveal the synergistic effect, the characterization of cone calorimeter was conducted to measure the important parameters, including the heat release rate (HRR), the total heat release (THR) and the residual mass.

The HHR results and ignition time of PP and its composites are presented in Figure 7a and Table 4, which were conducted by cone calorimeter at 50 kW/m^2^. Firstly, it is observed that, after introduction of AC, the ignition time of PP composites (AC-Ni-PP, AC-15Ni-Al-PP, AC-10Ni-Al-PP, AC-5Ni-Al-PP and AC-1Ni-Al-PP) is advanced more than NiO-PP and neat PP, resulting from the fact that AC promotes the cracking ability of PP fragment radicals into light hydrocarbons [32]. It is also observed that the neat PP shows the sharpest increase and decline for HRR curve, and the PHRR value reaches 1197 kW/m^2^. In comparison with pure PP, the PP modified by NiO or AC alone shows similar peak type with lower HRR values due to the carbonization reaction catalyzed by NiO. For the composites modified by AC, NiO or Al_2_O_3_ simultaneously, the combustion property suffers from significant change; the PHRR declines dramatically and the type of curves becomes gentle. The PHRR value reaches as low as 301 kW/m^2^ for AC-5Ni-Al-PP, reduced by 76% compared to neat PP. In summary, the results indicate that no significant influence occurred when the NiO or AC exists alone. However, the combination of the binary or ternary composites (NiO, AC or Al_2_O_3_) has a remarkable effect on the combustion property, indicating the presence of a synergistic effect, which promotes the flame retardancy.

The THR plots of PP and its composites are presented in Figure 7b. From the results, it is observed that the neat PP shows the highest THR value as much as 223 MJ/m^2^. In comparison with neat PP, the PP modified by NiO or AC alone shows a different degree of decline, reducing by 15.7 and 10.8%, respectively. For the composites modified by combined catalysts, further decrease in THR occurred. The decline in THR was attributed to the decrease in combustible products and increase in residual char. In addition, we also found that the PP modified by combined catalysts shows some difference. For the AC-5Ni-1Al-PP, it shows the lowest THR value (Table 4). The AC-1Ni-Al-PP and AC-Ni-PP are inferior to that of AC-5Ni-Al-PP. For the AC-10Ni-Al-PP and AC-15Ni-Al-PP, the decrease in THR value is the smallest among the composites modified by NiO, AC or Al_2_O_3_. The tendency observed in the THR is in good accordance with the MLR results. It is really interesting that the introduction of Al_2_O_3_ into the system exerts different effects. From the results, the THR value decreases first and then increases with increasing Al_2_O_3_ amount. We concluded that there are two reasons that result in the phenomenon. On the one hand, the introduction of Al_2_O_3_ could reduce the particle size of NiO and enhance the dispersion, which is proven by the XRD and TEM measurements [38,39]. Therefore, the increase in Al_2_O_3_ can promote the dehydrogenation and production of CNTs. On the other hand, the introduction of Al_2_O_3_ can react with NiO at high temperature, resulting in the production of nickel–aluminum spinel. In comparison with NiO, higher temperature is needed to reduce the nickel–aluminum spinel, leading to decrease in the NiO amount participating in the CNTs production [40,41]. Finally, the lowest THR value is achieved because of a compromise between dispersion and reduction difficulty of NiO.

The residual mass of PP and its composites with combustion time are presented in Figure 7c and Table 4. Overall, it is observed that the curves decline steeply between 100 and 300 s, while the residual mass decreases slowly at the later stage. This trend is inconsistent with the HRR result. Furthermore, it is also observed that the residual mass of AC-PP and NiO-PP is close to the initial addition of AC or NiO. In comparison with AC-PP or Ni-PP, the PP composites modified by the combined catalysts exhibited higher residual mass, especially for AC-5Ni-Al-PP, where the value reaches as high as 42%. It is concluded that carbon nanotubes are formed through several reaction steps. Firstly, during the combustion process, the long chain of PP is dehydrogenated into small molecules, including H_2_ [42,43], and then the NiO was reduced into metallic Ni by H_2_, CO, CH_4_ and so on. Subsequently, the small molecules, such as olefin, alkyne or other hydrocarbons, adsorbed on the surface of Ni and further decompose into carbon atoms [21,44,45]. Under the synergistic effect of AC, the carbon atoms were interconnected with each other along with Ni particle to produce dimmers and trimmers, etc. [23]. Eventually, dimmers and trimmers continue to react into five- to six-member rings and grow into CNTs. In addition, AC assists Ni species, catalyzing the dehydrogenation of PP fragment and growth of multielement ring into CNTs during combustion. Al_2_O_3_ not only promotes the crosslinking of the dehydrogenation products but also stabilizes the Ni particle, avoiding the movement and aggregation. Furthermore, AC and NiO also form a network-like structure, which favors the formation of a shield layer [25].

In summary, in comparison with neat PP, it is observed that the multicomponent modification composites all show lower heat release ratios and higher carbonization ability based on the results of HRR, THR and residual mass. It is well known that the pyrolytic flammable gas (evolved during combustion) can be reduced due to the barrier of char. When PP is heated, some volatile substances burn, and some substances dehydrogenate to form carbon nanotubes or other amorphous carbon. In turn, the carbon layer not only acts as a physical barrier hindering the transfer of heat and thermal decomposition products to the surface but also prevents the migration of oxygen from air. Especially for the multicomponent modification composites, the residual mass reaches as much as 40% under the synergistic effect of NiO, AC and Al_2_O_3_.

## 4. Conclusions

A simple and efficient method was successfully developed to synthesis a novel combined AC/NiO/Al_2_O_3_ catalyst. After introduction of the catalyst into PP, the thermal stability and flame retardancy of PP were dramatically improved. From the TG results, the PP modified by AC, NiO and Al_2_O_3_ shows that the maximum loss rate increased by 92.6 ℃ compared to the neat PP, revealing the excellent thermal stability. Imitative combustion experiments revealed that residual mass of modified PP reaches as much as 45%, while almost no residual char was observed for the neat PP. From the cone calorimetry, the PHRR and THRR of the multicomponent modification PP reduced by about 60 and 40% in comparison with the neat PP. From SEM and TEM study, the residual char exists in the form of CNTs, and the Ni particle is wrapped by the CNTs. With interaction between AC and Ni, the PP was dehydrogenated into light hydrocarbons and the carbon atoms grow into CNTs along with Ni particles. Al_2_O_3_ enhances the dispersion of NiO. Due to the synergistic effect of AC, NiO and Al_2_O_3_, a large amount of CNTs produced and acted as a physical insulating barrier, which prevents the feedback from the surface and transfer of oxygen and diminishes the release of flammable degradation products. Eventually, the PP composites modified by combined AC/NiO/Al_2_O_3_ catalyst exhibit higher decomposition temperatures and lower PHRR and THRR compared to neat PP.

## Figures and Tables

**Figure 1 polymers-15-02135-f001:**
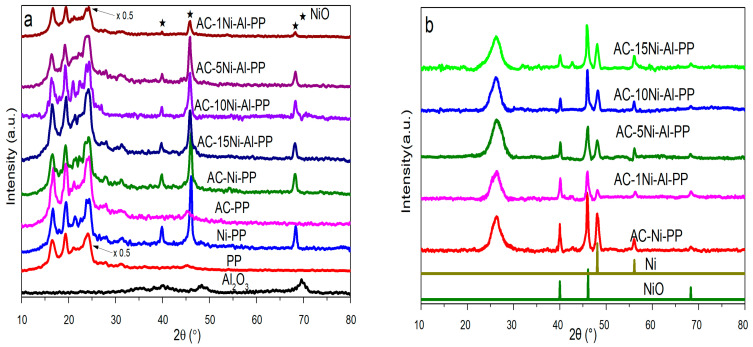
XRD patterns of (**a**) neat PP, PP composites and Al_2_O_3_ and (**b**) the combusted residues of PP composites after cone calorimeter tests.

**Figure 2 polymers-15-02135-f002:**
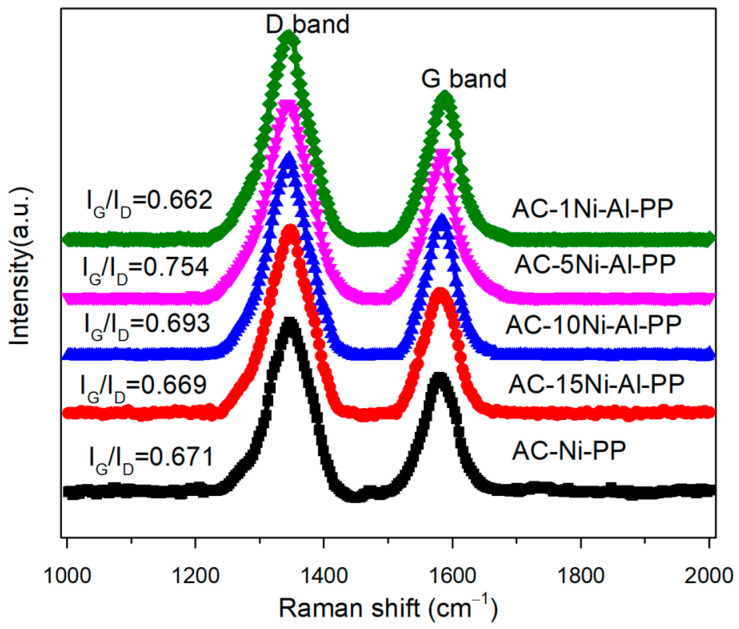
Raman spectra of PP composites after cone calorimeter tests.

**Figure 3 polymers-15-02135-f003:**
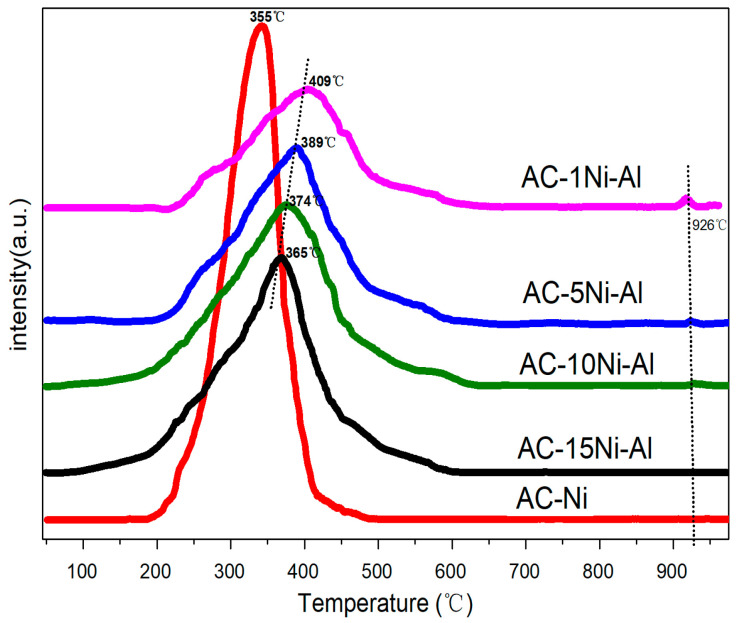
H_2_-TPR profiles of NiO and different ratios of AC, NiO and Al_2_O_3_ catalysts.

**Figure 4 polymers-15-02135-f004:**
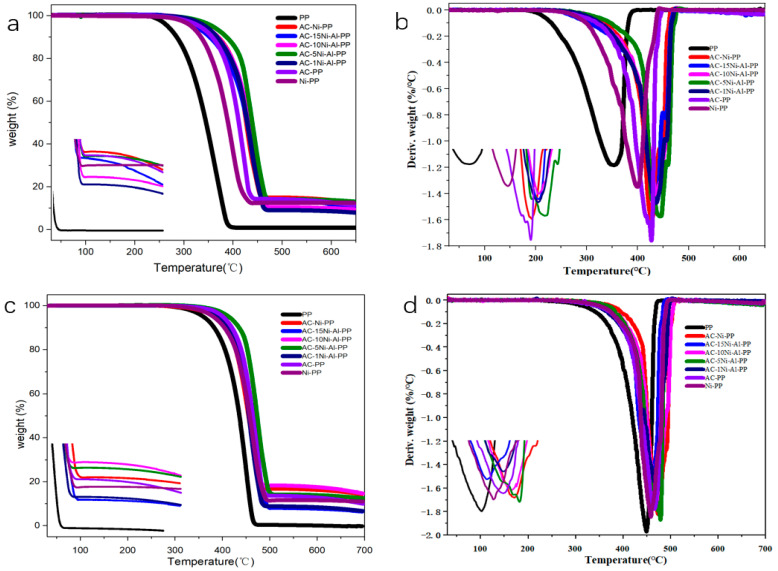
(**a**) TGA and (**b**) DTG curves of PP and its composites in air atmosphere. (**c**) TGA and (**d**) DTG curves of PP and its composites in nitrogen atmosphere.

**Figure 5 polymers-15-02135-f005:**
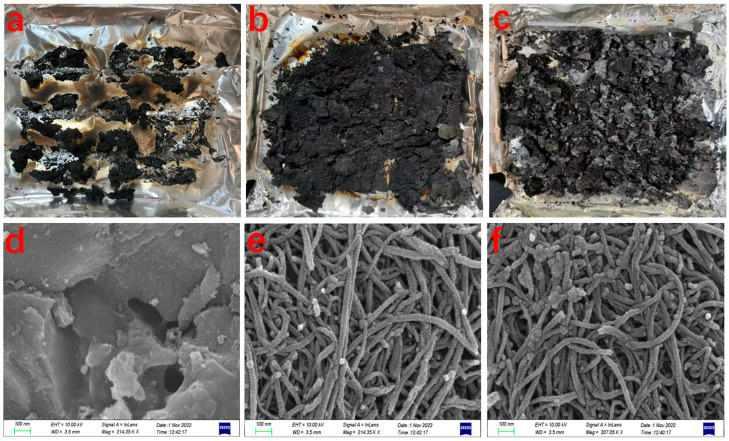
Digital photos of the residual char after the cone calorimeter tests from AC-PP (**a**), AC-Ni-PP (**b**), AC-5Ni-Al-PP (**c**); SEM images from AC-PP (**d**), AC-Ni-PP (**e**), AC-5Ni-Al-PP (**f**) after combustion at 700 °C.

**Figure 6 polymers-15-02135-f006:**
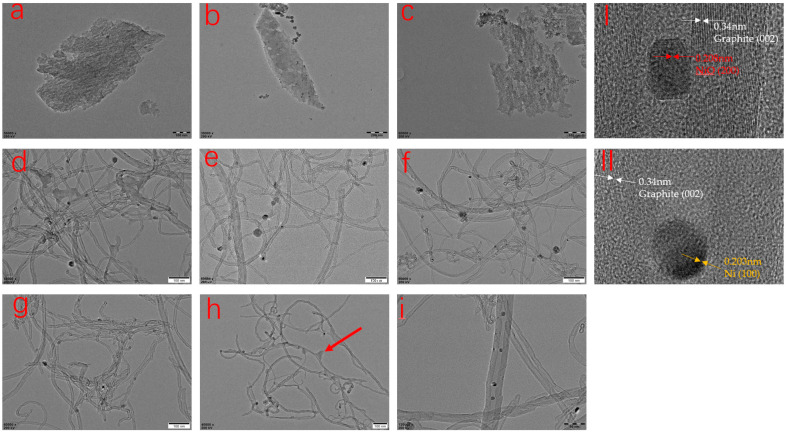
TEM images of the residual char from AC-PP (**a**), Ni-PP (**b**), Al-PP (**c**), AC-Ni-PP (**d**), AC-15Ni-Al-PP (**e**), AC-10Ni-Al-PP (**f**), AC-5Ni-Al-PP (**g**), AC-1Ni-Al-PP (**h**) and magnification image of AC-Ni-PP (**i**). (**I**) and (**II**) are HRTEM images of particles obtained from AC-Ni-PP, which prove the lattice spacing.

**Figure 7 polymers-15-02135-f007:**
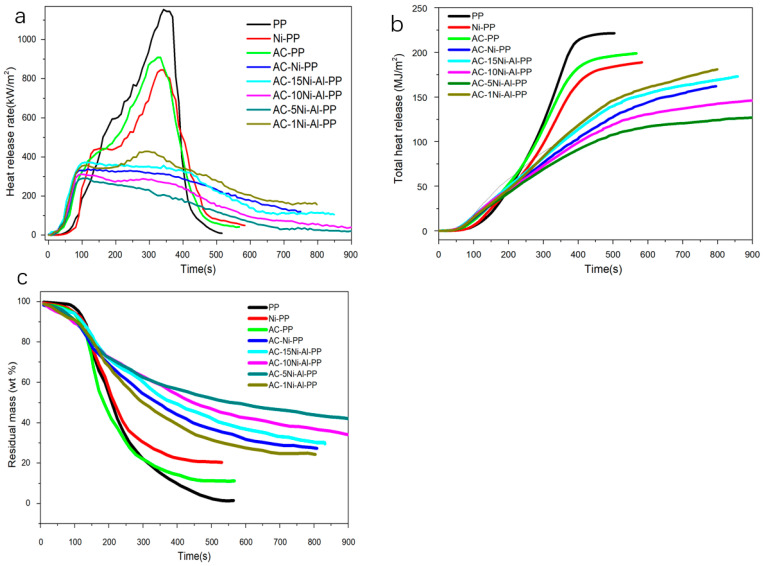
Heat release rate per unit area (**a**), total heat release (**b**) and residual mass of PP composites (**c**).

**Table 1 polymers-15-02135-t001:** Formulations of PP composites and the yield of residual chars.

Sample	PP (g)	AC (g)	NiO (g)	Al_2_O_3_ (g)	Sum (g)	Weight Ratio of NiO to Al_2_O_3_	Residual Chars
PP	100	0	0	0	100	-	0.17
AC-PP	85	15	0	0	100	-	9.71
Ni-PP	85	0	15	0	100	-	17.9
Al-PP	85	0	0	15	100	-	15.7
AC-Ni-PP	85	7.5	7.5	0	100	-	29.6
AC-15Ni-Al-PP	85	7.5	7.03	0.47	100	15:1	31.8
AC-10Ni-Al-PP	85	7.5	6.82	0.68	100	10:1	35.9
AC-5Ni-Al-PP	85	7.5	6.25	1.25	100	5:1	44.6
AC-1Ni-Al-PP	85	7.5	3.72	3.75	100	1:1	24.8

**Table 2 polymers-15-02135-t002:** Summary of TG and DTG data of PP and its composites in air atmosphere.

Samples	T_5_ ^a^ (°C)	T_10_ ^b^ (°C)	T_50_ ^c^ (°C)	T_max_ ^d^ (°C)
PP	267.1	285.2	345.1	352.1
AC-PP	331.9	357.3	411.9	427.5
Ni-PP	315.3	334.6	389.9	399.7
AC-Ni-PP	339.3	369.4	418.1	428.2
AC-15Ni-Al-PP	334.1	365.5	428.6	435.5
AC-10Ni-Al-PP	343.7	368.6	432.3	436.2
AC-5Ni-Al-PP	352.1	376.5	439.1	442.7
AC-1Ni-Al-PP	343.5	367.4	429.2	436.8

^a^ Temperature at 5% mass loss. ^b^ Temperature at 10% mass loss. ^c^ Temperature at 50% mass loss. ^d^ Temperature at maximum mass loss rate.

**Table 3 polymers-15-02135-t003:** Summary of TG and DTG data of PP and its composites in nitrogen atmosphere.

Samples	T_5_ ^a^ (°C)	T_10_ ^b^ (°C)	T_50_ ^c^ (°C)	T_max_ ^d^ (°C)
PP	359.7	382.5	435.8	449.2
AC-PP	401.1	420.0	464.1	465.5
Ni-PP	380.5	404.2	457.6	458.3
AC-Ni-PP	383.4	406.7	455.9	472.6
AC-15Ni-Al-PP	388.9	412.9	462.2	453.9
AC-10Ni-Al-PP	391.8	416.0	458.4	471.4
AC-5Ni-Al-PP	412.2	434.4	472.3	478.2
AC-1Ni-Al-PP	388.6	411.6	452.8	466.7

^a^ Temperature at 5% mass loss. ^b^ Temperature at 10% mass loss. ^c^ Temperature at 50% mass loss. ^d^ Temperature at maximum mass loss rate.

**Table 4 polymers-15-02135-t004:** Summary of the cone calorimetric results for PP and its composites.

Samples	t_i_ ^a^ (s)	PHRR (kW/m^2^)	THR (MJ/m^2^)	Residual Char (%)
PP	54	1151	223	1.2
AC-PP	9	911	198	11.4
Ni-PP	62	848	187	21.2
AC-Ni-PP	18	330	162	28.8
AC-15Ni-Al-PP	22	372	171	30.1
AC-10Ni-Al-PP	15	314	144	34.8
AC-5Ni-Al-PP	17	301	125	43.2
AC-1Ni-Al-PP	14	431	179	24.7

^a^ ignition time.

## Data Availability

The data presented in this study are available on request from the corresponding author.

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
