# Peer review of "Synergistic Effect of Activated Carbon, NiO and Al2O3 on Improving the Thermal Stability and Flame Retardancy of Polypropylene Composites"

_polymers, 2023, doi:10.3390/polym15092135_

Round 1

Reviewer 1 Report

Referee report:

Manuscript ID: polymers-2330729

Title: Synergistic effect of activated carbon, NiO, and Al2O3 on improving the thermal stability and flame retardency of polypropylene

Authors:   Mingqiang Shao, Ying Li, Yiram Shi, Jiangtao Liu, Baoxia Xue, and Mie Niu*

Summary: The work of Mei Niu and coworkers deals with enhancing the fire retardancy as well as the thermal stability of polypropylene composites. Using additives such as activated carbon, NiO,  and Al2O3, resulted in a synergistic activity that enhanced the fire retardancy and thermal stability by the in situ generation of carbon nanotubes.   The use of cone calorimetry for testing the heat release rate per unit area and the total heat released per unit area of the composite materials is particularly appealing.  However, there are some errors that need to be corrected prior to the publication of the work.  Owing to the originality and application the work of Mei Niu and coworkers is recommended for publication in “polymers” after major revision.

Major issues:   

1. English language need to be improved throughout the paper.

2.  Line 20, L20: Replace “AC-Ni-Al-PP” with “AC-5Ni-Al-PP”

3. L403: Replace “Ni and Ni particles” with “NiO”; Ni and NiO synonymously represented and Ni particles and this needs correction.

4. L325: In Figure 7(a), replace “heat release rate” with “heat release rate per unit area” or “heat flux”

5. L4: Replace “polypropylene” with “polypropylene composites”

Minor issues:

In addition the following minor changes can be attended for improving the quality of the paper.              

Line (L) number

Revision required

L9

Replace “promote” with “enhance”

L9

Add “preferential” before “complete”

L11

Replace “under” with “due to”

L11

Replace “According to the” with “from”

L14

Replace “compared with” with “compared to”

This change need to be applied throughout the paper

L18

Replace “Ni” with “NiO” 

Ni and NiO are not the same

L18

Replace “more” with “greater”

19

Replace “compared with” with “compared to”

20

Replace “oC” with “°C”

L20

Replace “AC-Ni-Al-PP” with “AC-5Ni-Al-PP”

L21

Add “per unit area” after “heat release rate”

L22

Add “per unit area” after “heat release”

L24

Add “composites” to the keywords

L29

Add “electrical appliances, automobiles, fibers, furniture and so on”  after “life”

L29

Delete “, including”

L33

Delete “which”

L34

Replace “widely” with “widespread”

L34

Replace “thereby” with “As a result”

L264

“Metal particles” to be replaced with either “Ni” or “NiO”depending on the context and the composite or the char

L264

Does “AC-Ni-PP” correspond to original composite or char? This needs classification.

L136-137

In Fig. 1, why “Ni metallic particles” are seen only in the combusted residues of the PP composites while “Ni metallic particles” are absent in the original “PP composite”

All through the manuscript the term “metal particles” is used both for “Ni and NiO”; Ni and NiO cannot be called as metal particles.  Ni is metal particle whereas NiO is metal oxide

L36-517

Owing to the constraint with time, the text in these lines could not be read line-by-line and the authors are requested to correct the text for any possible errors pertaining to either chemistry or English to avoid another revision.

Referee report:

Manuscript ID: polymers-2330729

Title: Synergistic effect of activated carbon, NiO, and Al2O3 on improving the thermal stability and flame retardency of polypropylene

Authors:   Mingqiang Shao, Ying Li, Yiram Shi, Jiangtao Liu, Baoxia Xue, and Mie Niu*

Summary: The work of Mei Niu and coworkers deals with enhancing the fire retardancy as well as the thermal stability of polypropylene composites. Using additives such as activated carbon, NiO,  and Al2O3, resulted in a synergistic activity that enhanced the fire retardancy and thermal stability by the in situ generation of carbon nanotubes.   The use of cone calorimetry for testing the heat release rate per unit area and the total heat released per unit area of the composite materials is particularly appealing.  However, there are some errors that need to be corrected prior to the publication of the work.  Owing to the originality and application the work of Mei Niu and coworkers is recommended for publication in “polymers” after major revision.

Major issues:   

1. English language need to be improved throughout the paper.

2.  Line 20, L20: Replace “AC-Ni-Al-PP” with “AC-5Ni-Al-PP”

3. L403: Replace “Ni and Ni particles” with “NiO”; Ni and NiO synonymously represented and Ni particles and this needs correction.

4. L325: In Figure 7(a), replace “heat release rate” with “heat release rate per unit area” or “heat flux”

5. L4: Replace “polypropylene” with “polypropylene composites”

Minor issues:

In addition the following minor changes can be attended for improving the quality of the paper.              

Line (L) number

Revision required

L9

Replace “promote” with “enhance”

L9

Add “preferential” before “complete”

L11

Replace “under” with “due to”

L11

Replace “According to the” with “from”

L14

Replace “compared with” with “compared to”

This change need to be applied throughout the paper

L18

Replace “Ni” with “NiO” 

Ni and NiO are not the same

L18

Replace “more” with “greater”

19

Replace “compared with” with “compared to”

20

Replace “oC” with “°C”

L20

Replace “AC-Ni-Al-PP” with “AC-5Ni-Al-PP”

L21

Add “per unit area” after “heat release rate”

L22

Add “per unit area” after “heat release”

L24

Add “composites” to the keywords

L29

Add “electrical appliances, automobiles, fibers, furniture and so on”  after “life”

L29

Delete “, including”

L33

Delete “which”

L34

Replace “widely” with “widespread”

L34

Replace “thereby” with “As a result”

L264

“Metal particles” to be replaced with either “Ni” or “NiO”depending on the context and the composite or the char

L264

Does “AC-Ni-PP” correspond to original composite or char? This needs classification.

L136-137

In Fig. 1, why “Ni metallic particles” are seen only in the combusted residues of the PP composites while “Ni metallic particles” are absent in the original “PP composite”

All through the manuscript the term “metal particles” is used both for “Ni and NiO”; Ni and NiO cannot be called as metal particles.  Ni is metal particle whereas NiO is metal oxide

L36-517

Owing to the constraint with time, the text in these lines could not be read line-by-line and the authors are requested to correct the text for any possible errors pertaining to either chemistry or English to avoid another revision.

Author Response

     Thank you very much for comments about our manuscript entitled “Synergistic effect of activated carbon, NiO and Al2O3 on improving the thermal stability and flame retardancy of polypropylene” submitted to Polymers.

     The comments are valuable and helpful for revising and improving our manuscript. We have browsed the comments carefully and made revision. We enclosed a revised manuscript according to the comments. The revised portion was marked in red in the manuscript.

    We are very sorry for our incorrect writing. According to the comments from you, we polished the manuscript with a professional assistance in writing, conscientiously. And the corrected sentences were rewriteen and marked in yellow in the manuscript.

     Furthermore, we want to answer one problem you proposed here that “why “Ni metallic particles” are seen only in the combusted residues of the PP composites while “Ni metallic particles” are absent in the original “PP composite”.

Response: In the original PP composite, NiO was produced by the calcination of Ni(NO3)2·6H2O. Therefore, the Ni metallic particle does not exist in the PP composites before combustion. However, methane (CH4), hydrogen (H2) or other reduction gases produced during combustion [1]. The CH4, H2 or other gases could reduce the NiO into Ni. Consequently, Ni metallic particles” are seen only in the combusted residues of the PP composites while “Ni metallic particles” are absent in the original “PP composite

References

[1] Y. Shen, W. Gong, B. Zheng, L. Gao, Ni–Al bimetallic catalysts for preparation of multiwalled carbon nanotubes from polypropylene: Influence of the ratio of Ni/Al, Appl. Catal. B 2016, 181, 769-778.

Reviewer 2 Report

In the study novel flame retardant to improve the thermal stability and flame retardancy of polypropylene was investigated. The authors evaluated the synergistic effect of activated carbon, NiO, and Al2O3. The issue is essential since polypropylene is widely used in many areas of life. However, some points must be improved before it is considered for publication.

1) Abstract: The authors should include more results to complement the scientific divulgation of the research.

2) 2.1. Materials: The authors must include more details about the materials used in experiments. For example, Polypropylene (PP: density, Melt Flow Rate (MFR), mass, Melt Flow Index (MFI),…)   

3) 2.3. Characterization, line 113: The text “Thermal gravimetric analyses (TGA)” should be replaced by Thermogravimetric Analysis (TGA) or Thermogravimetry (TGA).  

4) 3. Results and Discussion: Before the section “3.1. Composition analysis of PP composites before and after combustion”, the authors should include a paragraph to introduce the results and discussion. For example, The results are presented in the sections…

5) The quality of Figure 4 could be improved because, in the present form, the interpretation is very confusing.

6) Sections 3.3. and 3.5. must be revised because Figures 4 and 7 are presented between the text.

7) If possible, authors could include the mean ± standard deviation in the results. 

Author Response

Thank you very much for comments about our manuscript entitled “Synergistic effect of activated carbon, NiO and Al2O3 on improving the thermal stability and flame retardancy of polypropylene” submitted to Polymers.

The comments are valuable and helpful for revising and improving our manuscript. We have browsed the comments carefully and made. We enclosed a revised manuscript according to the comments. The revised portion was marked in green in the manuscript.

Round 2

Reviewer 2 Report

The manuscript has been improved and can be published in Polymers.